# Batch and Continuous Chromate and Zinc Sorption from Electroplating Effluents Using Biogenic Iron Precipitates

**Laura Castro [1,*], Fabiana Rocha [2], Jesús Ángel Muñoz [2], Felisa González [2] and María Luisa Blázquez [2]**

1 Department of Applied Mathematics, Materials Science and Engineering and Electronic Technology, School of Experimental Sciences and Technology, Rey Juan Carlos University, 28933 Móstoles, Spain

2 Department of Chemical and Materials Engineering, University Complutense of Madrid, 28040 Madrid, Spain; fabiroch@ucm.es (F.R.); jamunoz@quim.ucm.es (J.Á.M.); fgonzalezg@quim.ucm.es (F.G.); mlblazquez@quim.ucm.es (M.L.B.)

* Correspondence: laura.castro@urjc.es

**Abstract:** Nanoparticles of iron precipitates produced by a microbial consortium are a suitable adsorbent for metal removal from electroplating industry wastewaters. Biogenic iron precipitates were utilized as adsorbents for chromate and zinc in batch conditions. Furthermore, the iron precipitates were embedded in alginate beads for metal removal in fixed-bed columns, and their performance was evaluated in a continuous system by varying different operational parameters such as flow rate, bed height, and feeding system (down- and up-flows). The influence of different adsorption variables in the saturation time, the amount of adsorbed potentially toxic metals, and the column performance was investigated, and the shape of the breakthrough curves was analyzed. The optimal column performance was achieved by increasing bed height and by decreasing feed flow rate and inlet metal concentration. The up-flow system significantly improved the metal uptake, avoiding the preferential flow channels.

**Keywords:** chromate; wastewaters; adsorption; biogenic iron precipitates; alginate beads

## 1. Introduction

Industrial wastewaters from electroplating, mining, metal processing, or petroleum refineries contaminate the environment [1]. The management, treatment, and disposal of industrial wastewater containing potentially toxic metals is a growing worldwide concern for governments, policymakers, industries, and researchers [2]. Metals, such as chromium and zinc, are dangerous and toxic to the environment because they are non-biodegradable and accumulate in organisms through the food chain [3].

The most common technologies for water treatment are coagulation and flocculation, dissolved air flotation, sedimentation, filtration, ion exchange, reverse osmosis, and adsorption [4]. The development of sustainable materials and processes is gaining interest in many fields of research due to three major factors: (i) the environment; (ii) the economy; and (iii) the depletion of natural reserves.

Several adsorbents are involved in the removal or recovery of a great variety of contaminants, including metal ions, dyes, organics, and pharmaceuticals. Nanomaterials provide high specific surfaces in comparison to the limited active sites in conventional adsorbents, which leads to high adsorption kinetics and efficiency [5]. Metal oxide nanoparticles, such as alumina, titanium oxides, manganese, and iron oxides, exhibit interesting adsorptive properties and possess a potential application. Iron oxides are present in nature as different phases; however, the better-known adsorptive nanoparticles are the zero-valent iron (nZVI), $Fe_3O_4$ and $\gamma\text{-}Fe_2O_3$ [6,7]. These materials present different physicochemical properties due to the difference in their iron oxidation states and their capacity

for contaminant removal. The sorption behavior of iron precipitates involved in the removal of Cr (VI) from aqueous solution are summarized in Table 1.

The remarkable advantages of nanomaterials allow the development of clean, sustainable, and economic methods for environmental remediation. In this way, microbial iron precipitates can be a cost-effective source of bioadsorbents [8]. The surfaces of ferric iron minerals are positively charged at neutral pH because of their high points of net zero charge. Consequently, these biogenic materials are good adsorbents for species with negative charge such as phosphate ($PO_4^{3-}$) and bicarbonate ($HCO_3^-$) [9,10], and oxyanions of hazardous metals such as arsenate ($AsO_4^{3-}$), arsenite ($AsO_3^{3-}$) [11–13] or chromate ($CrO_4^{2-}$) [14]. In addition, natural organic matter from the negatively charged bacterial matrix binds strongly to ferric iron mineral surfaces. Cations are adsorbed onto iron precipitates through weak electrostatic interactions and/or hydrogen bonding [15].

**Table 1.** Summary of iron compounds used for Cr (VI) removal and adsorption capacity of Cr(VI).

| Iron Compounds | Adsorption Capacity (mg/g) | pH | Reference |
|---|---|---|---|
| Green rust | 55.01 | 9 | [16] |
| Ferrihydrite | 83.73 | 3 | [17] |
| Organo-Fe (III) composites | 51.8 | 3 | [18] |
| Micro-sized granular ferric oxide | 5.8 | 7 | [19] |
| Ferrihydrite | 60 | | |
| Siderite | 60 | 7.7 | [20] |
| Goethite | 20 | | |
| Nano iron oxide impregnated in chitosan bead | 69.8 | 5 | [21] |

Enrichment cultures containing *Geobacter* species grown anaerobically with synthetic ferrihydrite produce magnetite particles which, in contact with different cations, remove metal ions from solution according to the following order of high sorption efficiency: $Zn^{2+}$ > $Ni^{2+} \approx Co^{2+}$ > $Mn^{2+}$ [22].

Biogenic iron (oxyhydr)oxides may naturally diminish Cr (VI) concentrations through sorption and redox reactions that transform mobile Cr (VI) into less soluble Cr (III) species [23]. Cr (III) and Cr (VI) are adsorbed by biomineralized goethite at pH 5; Cr (VI) is adsorbed on nanometer goethite much easier than Cr (III) [24]. Schwertmannite ($Fe_8O_8(OH)_x(SO_4)_y$) produced by *Acidithiobacillus ferrooxidans* was an effective Cr (VI) adsorbent in continuous flow columns at a flow rate of 1 mL/min and pH 6 [25].

Few studies have been reported on hazardous metal removal from industrial wastewaters using biomaterials in continuous systems [26,27]. Further research development of the sorption process is needed at pilot and industrial scale for the treatment of real wastewaters.

The use of biopolymers as adsorbent support is another option for the application of micron/nano structures for wastewater treatment studies. Biomaterials provide a relatively inert response to aqueous systems within the matrix, and the high gel porosity enables high diffusion rates of pollutants [28]. Furthermore, the stabilization of the iron oxide particles through surface modification or addition of surface surfactant is required for its application in several fields [29]. Nevertheless, hybrid (inorganic–organic) materials have scarcely been investigated. Metal–organic frameworks/alginate composite beads have been used as adsorbents for Cr (VI) removal, creating more active sites. This material could also be regenerated after desorption experiments [30]. Magnetite coated with humic acid [31] and m-phenylenediamine [32] are effective materials to remove Cr (VI) from aqueous solution. Biochar and magnetite nanoparticle composites enhanced the adsorption and reduction of Cr (VI) from polluted waters at acidic to neutral pH as compared to biochar and magnetite nanoparticles separately [33].

In previous research, we demonstrated the ability of biogenic iron compounds to adsorb hazardous metals in synthetic wastewaters, especially the chromate and arsenate anions, under batch conditions [34]. The present study evaluates the chromate and zinc sorption efficiency of the iron precipitates produced by a natural microbial consortium using real electroplating industry wastewaters. In addition, the biogenic iron compounds were immobilized in alginate beads and hazardous metal sorption was studied in columns under different operating parameters: feed flow rate, bed height (amount of adsorbent) and the feeding system. The experimental data enabled determination of the optimum performance conditions that lead to the highest metal uptakes. This work is a preliminary step for the industrial application of nanometric iron precipitates in the decontamination of dilute wastewaters.

## 2. Materials and Methods

### 2.1. Biogenic Iron Precipitates

Iron compounds biosynthesized by an iron-reducing microbial consortium were used as adsorbents. The culture was grown using soluble ferric iron citrate (60 mM) as the iron source, and lactate (6 g/L) as the carbon source under anaerobic conditions for 3 days. The biogenic precipitates were identified as siderite ($FeCO_3$), magnetite ($Fe_3O_4$) and vivianite ($Fe_3(PO_4)_2 \cdot 8H_2O$). Overall, the surface area of the biogenic adsorbent was 56.98 $m^2$/g, the pore volume was 0.122 $cm^3$/g, and the average pore size was 83 Å [34].

### 2.2. Encapsulation of Iron Precipitates in Alginate Beads

The alginate beads were elaborated by dropping a 20 g/L sodium alginate aqueous solution through a syringe (internal diameter of 0.5 mm) into 3% $CaCl_2$ solution. The beads were rinsed with deionized water. The alginate beads containing the biogenic iron compounds were prepared by adding 20 g/L of the biogenic iron precipitates to the alginate solution.

### 2.3. Batch Adsorption Experiments

The adsorption tests were carried out with electroplating wastewaters from Industrial Goñabe (Valladolid, Spain). The electroplating wastewaters were characterized (pH 2.2; [Cr] = 2064 mg/L; [Zn] = 306.4 mg/L; [Fe] = 14.55 mg/L; [$Cl^-$] = 2760 mg/L; [$SO_4^{2-}$] = 52.6 mg/L, [B] = 3.08 mg/L). These measurements were performed by the company Hidrolab S.L. using ICP-MS (IT-AG-064, accredited methodology). The initial pH of the polluted solutions was adjusted with diluted HCl and with NaOH to different values (2.25, 4.0, 7.0 and 10.0).

Kinetic experiments were performed by adding 1 g/L of adsorbent to Erlenmeyer flasks containing 100 mL of diluted industrial wastewaters ([$Cr$]$_0$ = 10 mg/L) under stirring at room temperature (23 ± 1 °C). Sampling was performed at different time intervals (0, 15, 30, 60 and 120 min) and metal concentration was determined periodically.

The experiments were carried out in duplicates, and the average values were used for data processing.

### 2.4. Continuous Adsorption Experiments

Adsorption experiments were conducted at room temperature in small glass columns (2.5 cm inner diameter and 15 cm length) filled with biogenic iron precipitates encapsulated in alginate beads. The experiments were carried out with diluted solutions of electroplating effluents. The initial pH value of the metal solution was adjusted to pH = 4.0 with dilute NaOH, because it was evidenced that the maximum metal uptake was reached at this value in the batch experiments. Furthermore, the precipitation of the studied metals is limited at this pH. The wastewaters were fed from the top or from the bottom of the columns using a peristaltic pump (MasterFlex L/S (Vernon Hills, IL, USA)). In order to

provide a good distribution of the wastewaters and to prevent the loss of adsorbent, glass wool was positioned on the top and bottom of the column.

### 2.5. Analytical and Characterization Techniques

A pHmeter and ORPmeter Crison Basic 20 (sensitivity: 98%) were used to measure the pH and the redox potential of the samples, respectively.

Liquid samples were collected periodically, and chromium and zinc concentrations were determined by ICP-OES (Perkin Elmer Optima 2100 DV (Wellesley, MA, USA)).

Adsorbents (biogenic iron precipitates and beads) were coated with a thin layer of gold and observed in a scanning electron microscope (SEM) (JEOL JSM-6330 F (Tendo, Yamagata, Japan)).

Biogenic iron precipitates were characterized before and after the adsorption experiments using powder X-ray diffraction (XRD) on a Philips X'pert-MPD equipment with a Cu anode working at a wavelength of 1.5406 Å as the radiance source.

## 3. Results and Discussions

### 3.1. Batch Experiments

#### 3.1.1. Metal Sorption Using Biogenic Iron Precipitates

In a previous study, the adsorbent, iron oxide and siderite generated by a natural microbial consortium from an abandoned mine, was able to take up several potentially toxic metals from synthetic and dilute solutions. The highest metal uptake was obtained for chromate and arsenate at pH = 4, $q_{max}$ Cr = 0.14 mmol/g and $q_{max}$ As = 0.15 mmol/g, due to the electrostatic interaction between the metallic oxyanions and the positively charged surface of the biogenic iron compounds [34].

In this work, the biogenic iron precipitates were used as adsorbents with industrial wastewaters from a zincate process. The uptake of metallic ions on biogenic iron precipitates is strongly affected by the pH value of the solution. The pH had an effect on the chemistry of the metals and in the binding sites because of the protonation of functional groups [35]. The pH value of the electroplating wastewaters provided by Industrial Goñabe (Spain) was 2.25, and zinc and chromium were present in concentrations over the discharged legal limits (Table 1). Consequently, the adsorption of these metals using the biogenic iron compounds was considered in this study. Metal precipitation as a function of pH was also measured (Table 2). Chromium remained in solution at any pH, indicating that this metal was present as Cr (VI). At pH 4, iron was completely precipitated, and zinc concentration remarkably decreased over pH 7.

**Table 2.** Effect of the pH changes in the composition of the wastewaters generated in the electroplating process of Industrial Goñabe (each measurement was performed in triplicate with RSD <3%).

|  | pH 2.25 | pH 4 | pH 7 | pH 10 |
| --- | --- | --- | --- | --- |
| $[Cr]/mg·L^{-1}$ | 2064 | 1901 | 1827 | 1804 |
| $[Zn]/mg·L^{-1}$ | 306.4 | 269.1 | 14.9 | 0 |
| $[Fe]/mg·L^{-1}$ | 14.55 | 0 | 0 | 0 |

The initial chromium concentration for the sorption experiments was 10 mg/L (wastewater dilution was required) and the adsorption time was 2 h. The pH values of the study were 2.25, 4.0, 7.0 and 10. Figure 1 shows the chromium adsorption using the iron compounds biosynthesized by the natural consortium. The optimum pH for chromate adsorption was pH = 4. At this pH, $HCrO_4^-$ ions in solution are attracted by the surface of the biogenic particles with a net positive charge. At lower pH, $H_2CrO_4$ is the predominant species, and the interaction is weaker. At higher pH values, the anions in solution ($HCrO_4^-$ and $CrO_4^{2-}$) and the surface with net negative charge tend to be repelled.

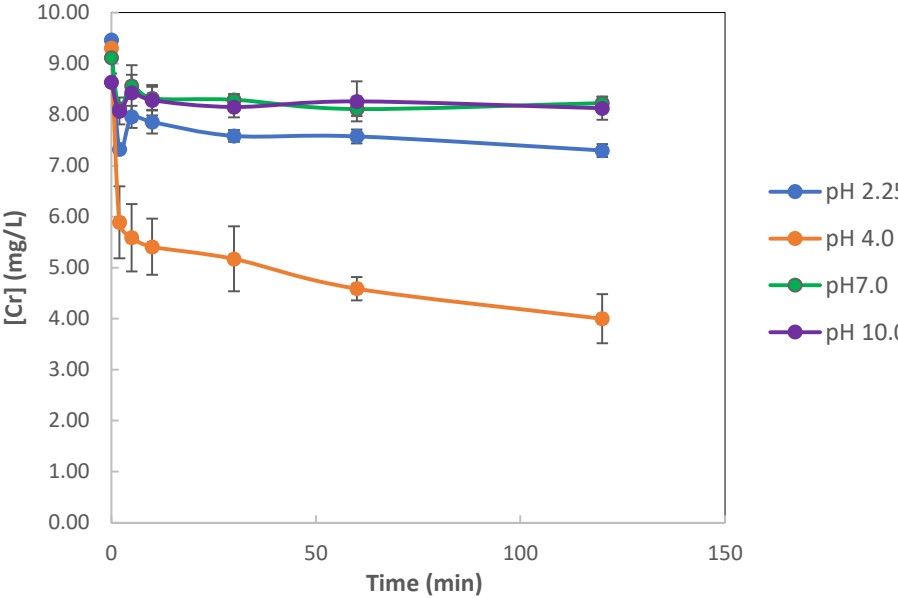

**Figure 1.** Effect of initial pH value on the metal uptake of chromate from real wastewaters by biogenic iron precipitates (1 g/L adsorbent).

Zinc cations from the electroplating wastewaters were not adsorbed on the biogenic iron nanoparticles (data not shown). Zinc cations in the aqueous phase were better adsorbed at high pH values than at very acidic pH because the organic residues of the bacterial matrix that coat the iron precipitates act as adsorbents [8,36]. At low pH values, lower than $pK_a$, the carboxylate groups in the biological molecules are protonated and the binding sites are blocked, preventing the access of metal ions through electrostatic repulsive forces. As shown in Table 2, most of the zinc ions were precipitated at circumneutral pH and were undetected at pH 10.

An increase in the pH value was observed after the adsorption process, indicating a proton consumption. In addition, the redox potential decreased, pointing out the reduction of chromate (Table 3).

**Table 3.** Variation of pH and redox potential in the adsorption experiments using the biogenic iron precipitates.

| $pH_{initial}$ | $pH_{final}$ | $E_{initial}$ (mV vs. Ag/AgCl) | $E_{final}$ (mV vs. Ag/AgCl) |
|---|---|---|---|
| 2.25 | 2.32 | 570 | 566 |
| 4.06 | 6.01 | 445 | 325 |
| 6.97 | 6.71 | 385 | 260 |
| 10.0 | 8.81 | 294 | 233 |

The iron-containing solids produced by the microbial consortium were mineralogically characterized by X-ray diffraction. Biogenic precipitates were identified as siderite ($FeCO_3$), magnetite ($Fe_3O_4$) and vivianite ($Fe_3(PO_4)_2 \cdot 8H_2O$) (Figure 2a). The residues were also analyzed after the adsorption experiments and chromium oxide and chromium phosphide had been detected (Figure 2b). Fe (II) ions present in the end products of the microbial metabolism could reduce Cr (VI). Adsorption was controlled by the active sites, then redox reactions took place between chromate ions and iron compounds containing Fe (II) (Equation 1).

$$CrO_4^{2-} \text{ (aq)} + 3Fe^{2+} \text{ (s)} + 7H^+ \rightarrow 3Fe^{3+} \text{ (aq)} + Cr^{3+} \text{ (aq)} + 4H_2O \tag{1}$$

Finally, the precipitation and complexation of Cr (III) occurred, and the final products were solid chromium complexes that covered the adsorbent.

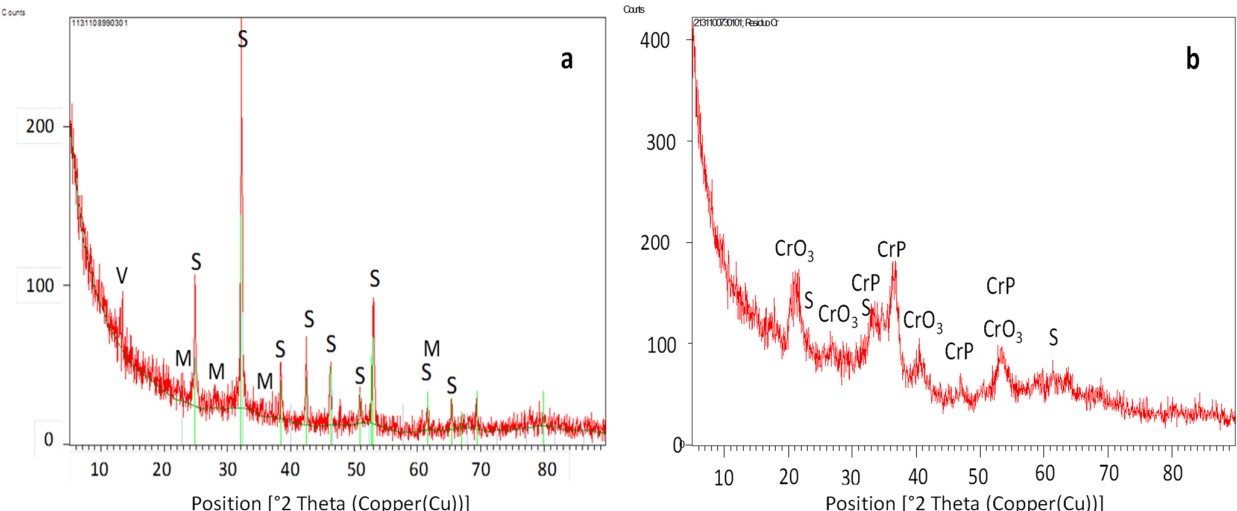

**Figure 2.** XRD patterns of (**a**) iron precipitates produced by the natural consortium and (**b**) iron precipitates after the adsorption experiments (V, vivianite; M, magnetite; S, siderite; $CrO_3$, chromium oxide; CrP, chromium phosphide).

The data from adsorption experiments were analyzed by three kinetic models (Lagergren pseudo first-order, pseudo second-order, and Elovich) to determine the metal sorption mechanism.

Lagergren's pseudo first-order model describes a process in liquid–solid phase systems where adsorption takes place on localized sites without interaction between adsorbed ions. This process obeys the following expression:

$$\frac{dq}{dt} = k_1(q_e - q) \tag{2}$$

where $q$ and $q_e$ are the adsorption capacities at time t and at equilibrium, respectively, and $k_1$ is the rate constant.

The pseudo second-order rate equation describes a kinetic process where the rate-limiting step is the chemical surface adsorption between the two phases. This model follows the equation:

$$\frac{dq}{dt} = k_2(q_e - q)^2 \tag{3}$$

where $k_2$ is the adsorption rate constant.

Elovich's model describes not only the adsorption of gas, but also aqueous ions onto solid phases. In this case, there are interactions between adsorbed ions on localized sites. The model expression is:

$$\frac{dq}{dt} = \beta e^{-\alpha q} \tag{4}$$

where $\alpha$ is the desorption constant and $\beta$ is the initial adsorption rate during the experiment.

Table 4 details the characteristic parameters of the kinetic models obtained by linear regression analysis. The experimental data are in a better concordance with the pseudo second-order model than with the first order or the Elovich model. These results indicate

that the rate-limiting step was chemical adsorption, due to the physicochemical interactions between adsorbent and adsorbate.

**Table 4.** Model kinetic parameters for the adsorption of chromium onto biogenic iron compounds generated by the microbial consortium at pH 4.

| Kinetic Model | Parameters | Cr (pH = 4) |
|---|---|---|
| | $q$ (mmol/g) | 0.153 |
| Pseudo first-order (Lagergren) | $k_1$ (min$^{-1}$) | 0.112 |
| | $R^2$ | 0.841 |
| | $q$ (mmol/g) | 0.103 |
| Pseudo second-order | $k_2$ (q/mmol·min) | 2.274 |
| | $R^2$ | 0.994 |
| | $\alpha$ (g/mmol) | 8.366 |
| Elovich | $\beta$ (mmol/g·min) | 120.48 |
| | $R^2$ | 0.920 |

3.1.2. Metal Adsorption Using Biogenic Iron Precipitates Encapsulated in Alginate Beads

Using biogenic iron precipitates as adsorbents is not easy due to their small size (Figure 3a,b). The separation of micron/nano materials from liquids is very difficult and expensive. Conventional separation methods are not efficient for these processes, and consequently, expensive separation techniques are required. Alginate was used to encapsulate the biogenic iron compounds, forming beads for wastewater treatment (Figure 3c,d).

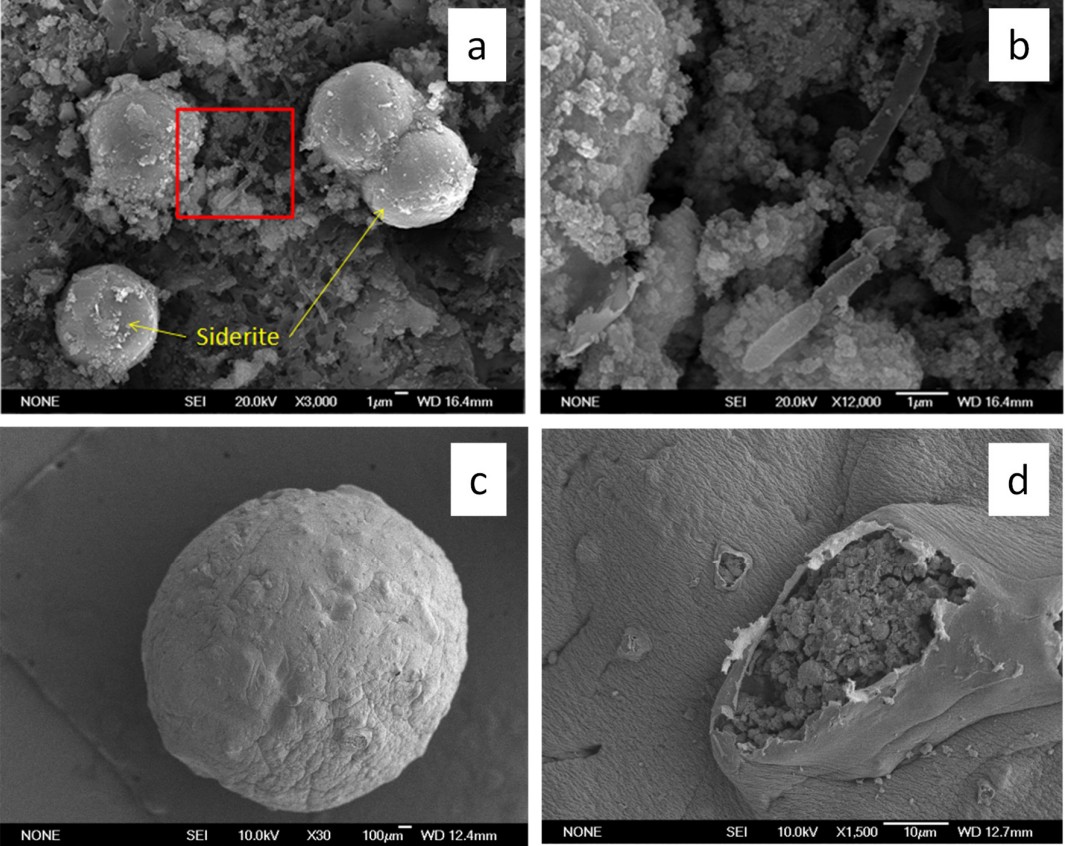

**Figure 3.** SEM images of (**a**) iron precipitates generated by the microbial consortium; (**b**) detail of the microorganisms and the surrounding particles; (**c**) alginate bead containing biogenic iron precipitates; and (**d**) a detail of the bead surface.

Batch experiments for the treatment of zincate wastewaters were performed with alginate beads with or without iron oxide particles at the optimum adsorption pH (pH = 4), as shown in Figure 4. Biogenic iron compounds alone were able to remove the chromate from the polluted solution while $Zn^{2+}$ remained in the wastewaters. Alginate beads without iron precipitates were able to remove zinc ions, while chromate ions were not adsorbed. Alginate is a linear copolymer of alternating blocks of D-mannuronic and L-guluronic acids with an abundance of carboxyl groups that play a key role in toxic metal adsorption. The pK_a value of carboxylic groups in the mannuronic acid is 3.38, and the pK_a of carboxylic groups in glucuronic acid is 3.65 [37]. At a pH value below the pK_a, the carboxylic groups are bound to hydrogen ions that limit the access of metallic cations due to repulsive forces. When pH increases and reaches a pH value of 4, carboxylates with negative charges are available and attract zinc cations, favoring metal uptake by alginate. Nevertheless, repulsive electrostatic forces are established between carboxylate groups and chromate anions.

Encapsulated iron precipitates showed a mixed behaviour, adsorbing both chromate and zinc ions. Carboxylate groups in the alginate gel that support the iron oxide particles interact with the zinc cations adsorbing them. The gel porosity favors the diffusion of chromate in the beads, allowing the interaction of anions with the surface of the iron solids with positive charge. The main drawback of the encapsulated iron compounds is that adsorption kinetics are slower than free biogenic iron compounds. Chromium concentration was 3.97 mg/L after 120 min in contact with iron particles, while chromium concentration was 7.49 mg/L after the same period of time in contact with the beads containing the biogenic iron precipitates.

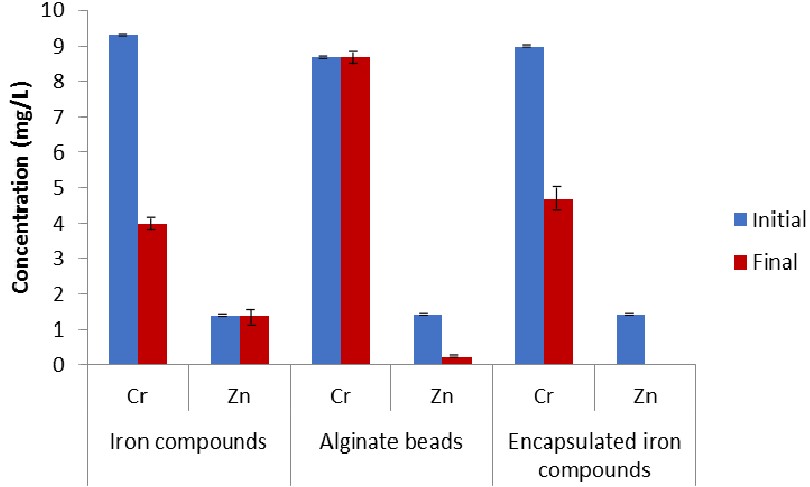

**Figure 4.** Comparison of the metal uptake of the adsorbents at pH = 4 and room temperature ([Adsorbent] = 1 g/L; contact time: 2 h with iron precipitates and alginate beads and 24 h with iron/alginate beads).

*3.2. Continuous Sorption*

The encapsulation of biogenic iron precipitates using biopolymers, such as alginate, is an alternative for the implementation of micron/nano adsorbents for continuous water treatment.

3.2.1. Influence of Flow Rate

Three columns with 10 cm bed height of encapsulated biogenic iron precipitates in alginate beads were tested at 2, 1 and 0.7 mL/min, and their column efficiency and metal uptake were compared (Figure 5). The increase in the feed flow rate allowed the treatment of higher amount of wastewater in less time; however, such increase led to a decrease in

the saturation time of the columns, indicating that the mass transfer zone became narrower.

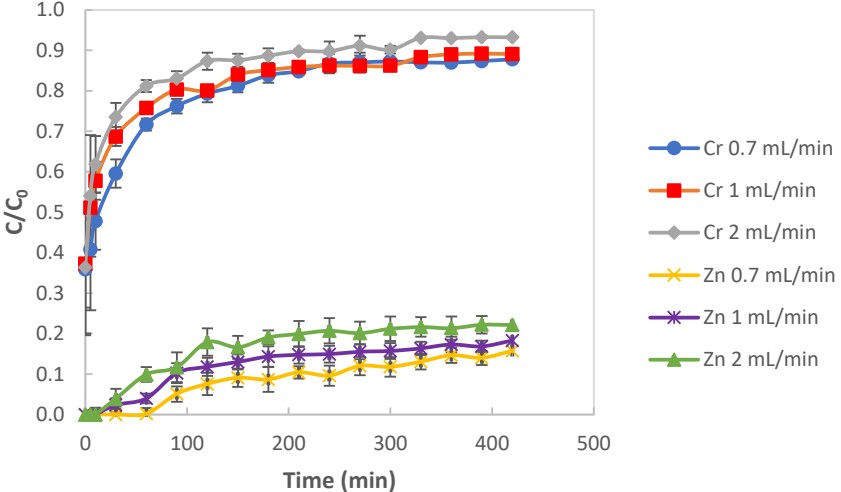

**Figure 5.** Chromate and zinc sorption breakthrough curves of drop-fed columns with alginate beads containing biogenic iron compounds using different feed flow rates (10 cm of bed height and 10 mg/L inlet Cr).

At a flow rate of 0.7 mL/min, the saturation time increased significantly because the slope of the breakthrough curve decreased, particularly in zinc uptake. The improvement of metal uptake was likely due to a longer contact time between the polluted solution and the beads. At 1 mL/min and 2 mL/min, the curves appear to almost be equal. This behavior was probably caused by the formation preferential flow channels due to the higher flow rate.

With respect to Cr uptake, the slopes of the curves seem to be very similar. This fact could be explained by the slow diffusion of chromate into the beads to interact with the iron precipitates with positively charged surfaces.

### 3.2.2. Influence of Bed Height

The effectiveness of continuous adsorption of two columns with 5 and 10 cm of bed height (10 mg/L Cr, 1 mL/min) was compared in Figure 6. The initial outlet concentration was affected by bed height and was lower for the column containing more beads. An increase in the bed height remarkably enhanced the saturation and service time of the columns because of the greater amount of adsorbent. Nevertheless, the column performance and the metal uptake decreased slightly, especially in the case of chromium. The lower Cr uptake could be caused by the slow diffusion of chromate into the beads.

Despite the amount of adsorbent increased, the uptake capacity was reduced. The reduction in metal adsorption with increasing bed height may be caused by a pressure drop along the column and by greater interferences between binding sites of the beads [38]. Furthermore, the efficiency of the higher bed column was lower likely due to the formation of preferential flow channels.

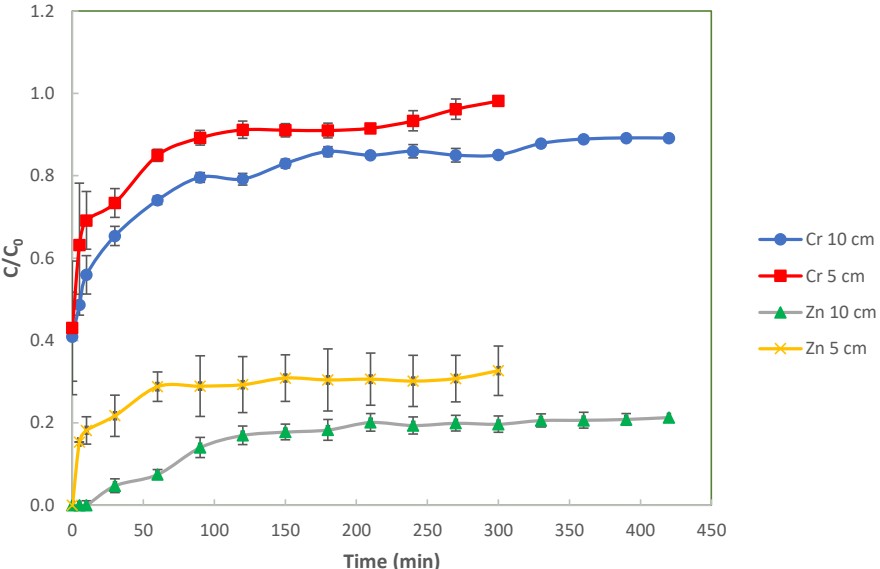

**Figure 6.** Chromate and zinc sorption breakthrough curves of drop-fed columns with alginate beads containing biogenic iron precipitates using different bed heights (10 mg/L inlet Cr and 1 mL/min flow rate).

### 3.2.3. Effect of Feeding System

The effect of the feeding systems was tested, drop- and reverse-fed, in columns with 10 cm bed height (10 mg/L Cr, 1 mL/min), as shown in Figure 7. Although saturation time was practically the same for chromium using both columns, higher metal uptake and column efficiency were reached in the reverse-fed system.

A remarkable difference between the feeding systems is that the shape of the breakthrough curve was favorably modified in the reverse feeding system. This curve showed a steeper slope and a lag period that increased the service time of the column.

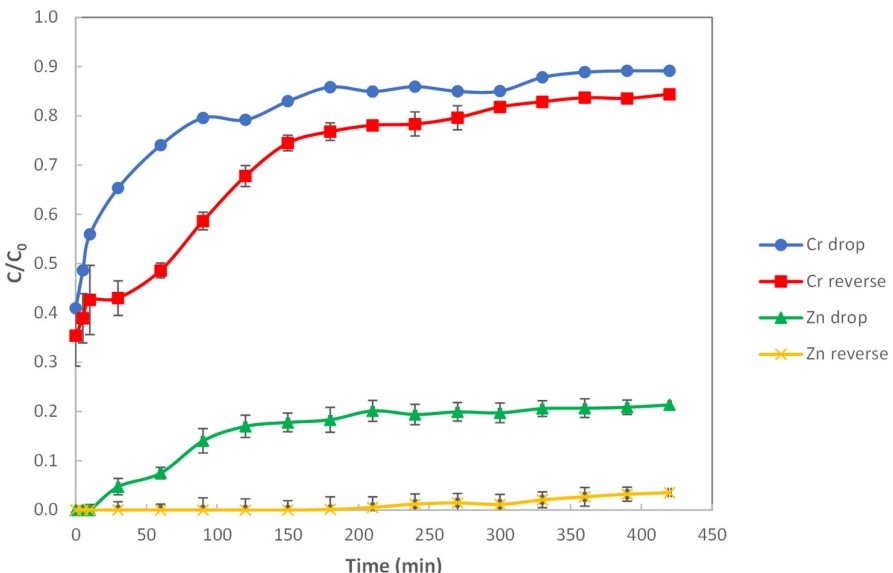

**Figure 7.** Chromate and zinc sorption breakthrough curves of columns with alginate bead-containing biogenic iron precipitates using down- and up-flow feeding systems (10 cm bed height, 10 mg/L inlet Cr, and 1 mL/min flow rate).

The reverse-fed column had a lower initial outlet concentration and reached a higher metal adsorption in the same period of time (309 μg of Cr were adsorbed after 120 min in the drop-fed column, compared to 509 μg of Cr in the reverse-fed column). The zinc concentration detected during the reverse-fed column performance was 0.05 mg/L after 420 min (521 μg of Zn were adsorbed after 420 min in the drop-fed column compared to 622 μg of Zn in the reverse-fed column). The improvement of the operation in the reverse feeding system seems to be linked to prevention of the formation of preferential flow channels.

The adsorbent used in these tests for potentially toxic metals removal is cheap; nevertheless, the possibility of reusing the alginate beads containing biogenic iron compounds in sorption–desorption cycles would be interesting from an economic point of view [39]. Diluted acids are commonly used as eluants with alginate. Sometimes there is a significant loss of effectiveness of the alginate after metal elution, and a regeneration step is needed [27]. Furthermore, the acidic pH could lead to the partial dissolution of iron particles and, consequently, a loss of the adsorbent mass.

### 4. Conclusions

A precipitate generated by a microbial consortium, a mixture of siderite ($FeCO_3$) and iron oxides, was used as adsorbent for the removal of potentially toxic metals. The optimal pH value for metal uptake was pH 4. Biogenic iron precipitates were able to remove the chromate from the polluted water; however, zinc remained in solution. Iron precipitates were encapsulated in alginate beads, and this material which adsorbed chromate and carboxylate groups in alginate was able to adsorb zinc cations.

The optimum conditions for metal adsorption with biogenic iron/alginate beads in column reactors were low feed flow rate (0.7 mL/min) and a reverse feeding system. High flow rates decreased the saturation time and narrowed the transfer zone of the columns. Regarding the amount of adsorbent, an increase in the bed height led to a greater pressure drop in the column and increased the saturation time. Column performance was significantly affected by the formation of preferential flow channels, which was avoided through an up-flow system. The breakthrough curve for chromium did not start at zero, indicating that longer contact time could improve the efficiency of the columns. Consequently, the biogenic precipitates used offer interesting opportunities for biotechnological applications in water treatment and the implementation in continuous systems, although improvements in the beads or in the design are necessary for full-scale applications.

**Author Contributions:** Conceptualization, J.Á.M.; Funding acquisition, M.L.B.; Investigation, L.C., F.R.; Methodology, J.Á.M.; Project administration, M.L.B.; Resources, F.G.; Validation, F.G., M.L.B.; Writing—review and editing, L.C. and J.Á.M. All authors have read and agreed to the published version of the manuscript.

**Funding:** This research was funded by the Spanish Ministry of Economy and Competitiveness (project MAT2014-59222R).

**Institutional Review Board Statement:** Not applicable.

**Informed Consent Statement:** Informed consent was obtained from all subjects involved in the study.

**Data Availability Statement:** Not applicable.

**Acknowledgments:** The authors are grateful for the financial support given by the Spanish Ministry of Economy and Competitiveness to fund this work (project MAT2014-59222R).

**Conflicts of Interest:** The authors declare no conflict of interest.

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
