# Peer review of "Batch and Continuous Chromate and Zinc Sorption from Electroplating Effluents Using Biogenic Iron Precipitates"

_minerals, doi:10.3390/min11040349_

Round 1
Reviewer 1 Report
Review of manuscript: minerals-1134961
This script describes the sorption of chromate and zinc from electroplating effluent using biogenic iron compounds, and discussion for the effective continuous removal using column. It is an interesting script, but it is not written properly. There are some points, which require major revision and need to be clarified in the revised text. The points are described below.
- 1 Adsorbent: You should write how to prepare the biogenic iron compounds. Also, you should add the XRD pattern of the adsorbent.
- Materials and Method, l. 95-99: This part should be independent. You should create “2.3 Alginate beads” and this part move to this section. Also, l. 98, “The alginate beads containing ,,,,the alginate solution.” Is moved into l. 96.
- Materials and Methods: l. 121, “Analytical and characterization techniques” is deleted. No enters in l. 120-l. 125. l. 124, Adsorbent means alginate beads or biogenic iron powder? You should write more clearly.
- Materials and Methods: You should add the chemical composition and pH of the industrial wastewater used in this study.
- Figure 1: During this experiment, the pHs of the solution are constant? You should write more information for this experimental condition.
- Figure 2: You should add the biogenic iron compound before forming beads.
- Figures: You should revise these figures for tick marks, range (from zero), frame and so on.
- 2.1 Effect of feed flow rate: You should indicate other flow rates, 1 mL/min and 2 mL/min, and should be compared.
- 2.2 Effect of bed hight: l. 226-227, “the metal uptake decreased slightly,” with increasing bed height? You should write more clearly.
- General: You should add the XRD patterns of the beads before and after removal test. Also, the elution of iron and alginate from the beads should be examined. This beads can be reused for removal or not? It is very important point for this study.
I recommended publication of this paper, subject to the above points being satisfactorily addressed.
Author Response
- 1 Adsorbent: You should write how to prepare the biogenic iron compounds. Also, you should add the XRD pattern of the adsorbent.
Author’s response:
The microbial cultivation has been now briefly included in the manuscript. This section includes a reference where the XRD pattern is published.
- Materials and Method, l. 95-99: This part should be independent. You should create “2.3 Alginate beads” and this part move to this section. Also, l. 98, “The alginate beads containing ,,,,the alginate solution.” Is moved into l. 96.
Author’s response:
Now we have created a new section to describe the alginate fabrication following the reviewer’s recommendation.
- Materials and Methods: l. 121, “Analytical and characterization techniques” is deleted. No enters in l. 120-l. 125. l. 124, Adsorbent means alginate beads or biogenic iron powder? You should write more clearly.
Author’s response:
We have corrected the mistake. Furthermore, we have clarified the term “adsorbent” and we have included the iron compounds and the beads following the reviewer’s suggestions.
- Materials and Methods: You should add the chemical composition and pH of the industrial wastewater used in this study.
Author’s response:
The required information has been now included in the Materials and Methods section.
- Figure 1: During this experiment, the pHs of the solution are constant? You should write more information for this experimental condition.
Author’s response:
The variation of pH and redox potential is now included in the manuscript. The pH value increased and the potential decreased due to the chromium reduction and precipitation.
- Figure 2: You should add the biogenic iron compound before forming beads.
Author’s response:
Following the reviewer’s suggestion, we have included the SEM images of the biogenic iron compounds.
- Figures: You should revise these figures for tick marks, range (from zero), frame and so on.
Author’s response:
Figures have been revised and improved.
- 1 Effect of feed flow rate: You should indicate other flow rates, 1 mL/min and 2 mL/min, and should be compared.
Author’s response:
This section has been improved following the reviewer’s recommendation.
2 Effect of bed hight: l. 226-227, “the metal uptake decreased slightly,” with increasing bed height? You should write more clearly.
Author’s response:
We have tried to clarify the sentence in the manuscript. Despite of the amount of adsorbent increased, the uptake capacity was reduced.
- General: You should add the XRD patterns of the beads before and after removal test. Also, the elution of iron and alginate from the beads should be examined. This beads can be reused for removal or not? It is very important point for this study.
Author’s response:
XRD patterns of the biogenic iron compounds before and after the removal experiments have been included following the reviewer’s suggestion.
The possibility of reusing the adsorbents was not the objective of this work and the material is very cheap. Generally, the elution with diluted acids leads to a loss of binding capacity in alginates. Furthermore, part of the iron could be dissolved at very low pH.
Reviewer 2 Report
The authors report their results about chromate and zinc sorption under batch and continuous conditions from effluents. A related sorbent material was prepared, characterized earlier and tested under batch conditions (see ref. 25). In the present study, this material was embedded into alginate beads. Major findings are appropriately summarized in Conclusions.
1. In introduction, 25 related references are cited, many of them from the early 2000’s. The authors do not appear to be aware of recent important, relevant reviews. Examples for Cr removal: J. Environ. Management, 2021, 280, 111809 and 2021, 279, 111547; Int. J. Environ. Res. Public Health 2020, 17, 543.
Furthermore, examples for the use of alginate-based materials is also known. See, for example: Chemosphere, 2021, 270, 129487. This section of the manuscript should be more focused!
2. Another major hiatus of the ms is the lack of instrumental characterization of the samples with the exception of SEM (Fig. 2). This means, among others, the lack of information about the adsorbents before and after their use in adsorption studies. Furthermore, if one is concerned about the environment, to have information of the reuse of the adsorbents should also be a key question.
3. Additional remarks
i) page 6, lines 216/217: “At a flow rate of 0.7 mL/min, the saturation time increased considerably while the slope of the breakthrough curve decreased, particularly in zinc uptake.” The latter statement is true; however, with respect to Cr uptake, the slope of the 1 mL/min and 0.7 mL/min curves appears to be almost identical.
ii). page 7, section 3.2.2: the effect of bed height is very significant for Zn in comparison to that of Cr – add an explanation.
iii) page 8
- in the paragraph below Fig. 6, data with respect to Cr adsorption using drop-fed vs reverse-fed column are given; similar data for Zn are missing;
- In the next paragraph it is stated that the “optimum conditions for metal adsorption…were low feed flow rate (0.7 mL/min).” Why is it then that data shown in Fig. 6 were collected with a flow rate of 1.0 mL/min?
iv) It would be useful to include a table with recent literature results. This would give readers a possibility to compare the results of this study with those reported previously.
Author Response
- In introduction, 25 related references are cited, many of them from the early 2000’s. The authors do not appear to be aware of recent important, relevant reviews. Examples for Cr removal: J. Environ. Management, 2021, 280, 111809 and 2021, 279, 111547; Int. J. Environ. Res. Public Health 2020, 17, 543.
Furthermore, examples for the use of alginate-based materials is also known. See, for example: Chemosphere, 2021, 270, 129487. This section of the manuscript should be more focused!
Author’s response
We thank the reviewer’s suggestion. We have included more recent references to improve the introduction.
- Another major hiatus of the ms is the lack of instrumental characterization of the samples with the exception of SEM (Fig. 2). This means, among others, the lack of information about the adsorbents before and after their use in adsorption studies. Furthermore, if one is concerned about the environment, to have information of the reuse of the adsorbents should also be a key question.
Author’s response:
XRD patterns of the biogenic iron compounds before and after the removal experiments have been also included following the reviewer’s suggestion.
The possibility of reusing the adsorbents was not the objective of this work and the material is very cheap. Generally, the elution with diluted acids leads to a loss of binding capacity in alginates. Furthermore, part of the iron could be dissolved at very low pH.
- Additional remarks
- i) page 6, lines 216/217: “At a flow rate of 0.7 mL/min, the saturation time increased considerably while the slope of the breakthrough curve decreased, particularly in zinc uptake.” The latter statement is true; however, with respect to Cr uptake, the slope of the 1 mL/min and 0.7 mL/min curves appears to be almost identical.
Author’s response
The section 3.2.1. has been improved following the reviewer’s comment.
ii). page 7, section 3.2.2: the effect of bed height is very significant for Zn in comparison to that of Cr – add an explanation.
Author’s response
The effect of different variables in Cr uptake is less significant than in Zn adsorption due to the slow diffusion of chromate into the beads to interact with the iron compounds with positively charged surfaces. We have clarified this point in the manuscript.
iii) page 8
- in the paragraph below Fig. 6, data with respect to Cr adsorption using drop-fed vs reverse-fed column are given; similar data for Zn are missing;
Author’s response
Data for Zn adsorption has been now included in the manuscript following the reviewer’s suggestion.
- In the next paragraph it is stated that the “optimum conditions for metal adsorption…were low feed flow rate (0.7 mL/min).” Why is it then that data shown in Fig. 6 were collected with a flow rate of 1.0 mL/min?
Author’s response
Figure 6 shows the effect of the feeding system and we set the feed flow rate to 1 mL/min. This rate was set also to analyze the effect of the bed height. When the effect of the feed flow rate in the metal uptake was studied, different rates were compared and it was observed that lower feed flow rates (0.7 mL/min) led to higher yields.
- iv) It would be useful to include a table with recent literature results. This would give readers a possibility to compare the results of this study with those reported previously.
Author’s response
Thank you for the suggestion. We have included a table summarizing examples.
Reviewer 3 Report
The article contains very interesting and practically significant results. The influence of the main factors on the process of Cr and Zn removing from solutions has been investigated. The experiment is described in sufficient detail and the selected conditions are justified.
But I have a few questions and wishes for the authors:
- the results of kinetics study can be processed using kinetic models, hereof the mechanism of sorption can be suggested and the rate constant can be calculated;
- in Figures 1 and 3 confidence intervals are required for concentration;
- in Figure 1 at pH = 2.25, 7 and 10 at 15 min there is a sharp decrease in Cr concentration, which is absent on the curve for pH = 4. How can this be explained? Or is this behavior within the measurement error?
- I think, that effect of feed flow rate on metal uptake for Cr is insignificant;
- why the process time for two columns with 5 and 10 cm is different (Figure 5)?
Author Response
- the results of kinetics study can be processed using kinetic models, hereof the mechanism of sorption can be suggested and the rate constant can be calculated;
Author’s response:
Different kinetic models (Lagergren pseudo first-order, pseudo second-order and Elovich) have been evaluated following the reviewer’s recommendation. The different constants have been calculated and a mechanism has been proposed.
- in Figures 1 and 3 confidence intervals are required for concentration;
Author’s response:
Figures have been improved following the reviewer’s recommendation.
- in Figure 1 at pH = 2.25, 7 and 10 at 15 min there is a sharp decrease in Cr concentration, which is absent on the curve for pH = 4. How can this be explained? Or is this behavior within the measurement error?
Author’s response:
The behavior is due to a fast adsorption followed by a desorption caused by the interaction between ions. At pH 4, the sharp decrease in Cr concentration is not observed or it is softer. Probably there are less repulsive forces between ions on the surface.
- I think, that effect of feed flow rate on metal uptake for Cr is insignificant;
Author’s response:
The section 3.2.1. has been improved. The effect of feed flow rate on Cr uptake is insignificant probably due to the slow diffusion of chromate into the beads to interact with the iron compounds with positively charged surfaces.
- why the process time for two columns with 5 and 10 cm is different (Figure 5)?
Author’s response:
Authors decided sampling during longer time because the column filled with 10 cm contained more adsorbent and more time could be required to define the breakthrough curve.
Reviewer 4 Report
It is well written brief paper with well approximable analytical curves, but it is not extremely novel result.
Despite this fact, it is obvious that such "data paper" can be published in present form or with minor revisions.
Author Response
It is well written brief paper with well approximable analytical curves, but it is not extremely novel result.
Despite this fact, it is obvious that such "data paper" can be published in present form or with minor revisions
Author’s response
We would like to thank the reviewer’s consideration. The main novelty of the study is the application of the biogenic iron compounds for the water treatment in a continuous system with real effluents. There are few works using biogenic iron compounds and the experiments are usually performed with synthetic solutions.
Round 2
Reviewer 1 Report
A few points should be revised as follows;
- You should write the chemical composition of industrial wastewater used in this study. Are there only Cr, Zn and Fe ions? No Na+, Cl-, SO42- or something? You should write more clearly.
- Graphs should be improved. Tick marks should be inward, and should be square framework.
Author Response
Reviewer 1
A few points should be revised as follows;
- You should write the chemical composition of industrial wastewater used in this study. Are there only Cr, Zn and Fe ions? No Na+, Cl-, SO42- or something? You should write more clearly.
Authors response:
These wastewaters contain different ions and elements that we do not have included ([Cl-] = 2760 mg/L; [SO42-] = 52.6 mg/L, [B] = 3.08 mg/L…). In the study, we have considered the most abundant and toxic heavy metals. We have included more data following the reviewer’s suggestion.
- Graphs should be improved. Tick marks should be inward, and should be square framework.
Authors response:
We have modified the graphs following the reviewer’s comment.
Reviewer 2 Report
In general, I am satisfied with the modified manuscript.
With respect to my remark in entry 2 with respect to information of the reuse of the adsorbents, the response is “The possibility of reusing the adsorbents was not the objective of this work and the material is very cheap. Generally, the elution with diluted acids leads to a loss of binding capacity in alginates. Furthermore, part of the iron could be dissolved at very low pH.”
I can hardly accept the first part of the remark (…not the objective of this work). With respect to the following arguments, these should be added to the ms. In this way, readers can clearly seen that this material is an unstable preparation and its practical application is highly questionable.
Author Response
Reviewer 2
With respect to my remark in entry 2 with respect to information of the reuse of the adsorbents, the response is “The possibility of reusing the adsorbents was not the objective of this work and the material is very cheap. Generally, the elution with diluted acids leads to a loss of binding capacity in alginates. Furthermore, part of the iron could be dissolved at very low pH.”
I can hardly accept the first part of the remark (…not the objective of this work). With respect to the following arguments, these should be added to the ms. In this way, readers can clearly seen that this material is an unstable preparation and its practical application is highly questionable.
Author’s response:
Thank you for your comment. We have included a paragraph about elution following the reviewer’s suggestion.